# Anterior and Apical Prolapse: Comparison of Vaginal Mesh Surgery to Vaginal Surgery with No Mesh

**DOI:** 10.3390/jcm12062212

**Published:** 2023-03-13

**Authors:** Charlotte Cassagne, Francoise Perriard, Arnaud Cornille, Jennifer Salerno, Laure Panel

**Affiliations:** 1Department of Gynecology, Clinique Beau Soleil, 34070 Montpellier, France; 2Equipe de Recherche AESIO SANTE, Clinique Beau Soleil, 34070 Montpellier, France

**Keywords:** anterior sacrospinous suspension, pelvic organ prolapse, vaginal surgery, native tissue repair, cystocele, vaginal mesh

## Abstract

Aim of the study: The aim of this study was to evaluate the anatomical results after an anterior sacrospinous ligament fixation (ASSLF) with native tissue repair (anterior colporraphy and apical suspension with prolene) compared to mesh repair for the correction of anterior prolapse at 12 months after surgery. Materials and methods: A monocentric prospective study comparing two similar cohorts who underwent ASSLF was conducted. The primary endpoint was the gain in the position of the Ba point relative to its position before surgery and twelve months after surgery. The secondary endpoints consisted of objective results, which were assessed using validated questionnaires. Results: Fifty-three women were included in the native tissue repair group between June 2019 and March 2020. They were compared to 53 women operated on with anterior and apical mesh. There was no difference with respect to the Ba point after 1 year between the two groups (−2 [−3; 1.5]; −2 [−3; 1], *p* = 0.9789). The apex was significantly better corrected in the native tissue repair group (−7 vs. −6, *p* = 0.0007). There was also a better correction on the rectocele in the native tissue repair group (−3 vs. −2, *p* = 0.0178). The rate of Stage 2 anterior vaginal prolapse at one year was approximately 30% in both groups (no statistical difference). Conclusions: ASSFL without mesh does not increase the risk of cystocele recurrence at 1 year after surgery. A future prospective comparison of this native tissue repair technique with mesh suspension is necessary to explore these preliminary findings.

## 1. Introduction

Pelvic organ prolapse (POP) is a common condition in women with an estimated prevalence of 30–50% (all stages of prolapse) [1]. Urinary symptoms, digestive symptoms, or sexual functional disorders may also be associated, which can lead to an alteration in body image and quality of life.

The treatment of prolapse can be conservative (perineal rehabilitation or the use of pessaries) or surgical (mostly 80–90% by vaginal route) [2]. A high grade of anterior prolapse is often associated with an apical defect.

Surgery is the most common type of intervention to treat vaginal wall prolapse, and there are a large number of repair techniques. In 1968, Richter described the sacrospinous ligament fixation (SSLF) technique, which allows for the suspension of the vaginal apex at the sacrospinous ligament after a direct posterior visual approach [3]. This posterior surgical approach carries some risks, such as pudendal nerve injury, vascular injury, chronic pain, dyspareunia, and recurrence rate of 30% for cystocele [4].

Before 2019, the most commonly performed vaginal procedures for anterior and apical POP surgical correction in France were mesh surgeries. This native tissue repair technique reduced the risk of prolapse recurrence and had a shorter operating time and cost benefits [5,6,7]. However, after evaluations of mesh-related complications (vaginal mesh exposure, chronic pain, and dyspareunia), the FDA decided to stop the sale and production of implantable devices marketed for vaginal prolapse surgery in the USA [8,9]. Similarly, the French authorities ceased marketing these implantable devices, and vaginal mesh repair procedures are no longer performed in France.

Since 2019, surgeons practicing in countries where vaginal mesh repairs are no longer an option have been worried about the complications (cystocele failure or recurrence) associated with reverting to the vaginal approach for high-grade cystoceles and apical defects. In fact, the Cochrane meta-analysis [10] (2016) demonstrated better anatomical outcomes after mesh surgery versus surgery without mesh for cystoceles and no difference in functional outcomes. This meta-analysis included many studies with large, wide, anterior native tissue repair techniques. The variety of techniques and the absence of a standardization of management make it difficult to evaluate them through high-level evidence studies.

Modified sacrospinous ligament fixation by the anterior approach after the dissection of the paravesical space has been described [11]. Several studies have evaluated its feasibility and anatomical effectiveness by showing a longer vagina, a more proximal vaginal apex, and less cystocele than with suspension at the sacrospinous ligament by the posterior approach [12].

To date, there is little data on anterior sacrospinous ligament fixation and the comparison between mesh and meshless surgery to treat anterior and apical prolapses in the same procedure. We also propose to conduct a prospective historical study comparing native tissue repair versus mesh repair during anterior sacrospinous ligament fixation.

Our primary hypothesis is based on the results of El Nazher’s study [6] on anterior repair without apical suspension. This study showed an improvement of the Ba point at 1 year after surgery in favor of mesh repair when compared to anterior colporrhaphy (Mean −2.7 ± 0.7 vs. −1.8 ± 1.4, *p* < 0.001). Therefore, we seek to demonstrate this variation of 1 cm for the anterior anatomical result compared with mesh repair for cystocele when performed at the same time as ASSLF.

In other words, the aim of this study is to evaluate whether native tissue repair for anterior and apical prolapse is less efficient based on the anatomical result at one year after surgery compared to an anterior vaginal mesh suspension.

## 2. Materials and Methods

### 2.1. Design

We performed a monocentric prospective study comparing two similar cohorts of women who had an anterior sacrospinous ligament fixation. The study was conducted at the Beau Soleil clinic, a referral urogynecology center in France.

This trial was approved by the Human Research Ethics Committee (NORD-OUEST 1 n°22.01387.000079—25/05/2022) and registered under the number ID-RCB 2022-A00414-39.

### 2.2. Participants

The study population consisted of women older than 18 years old with pelvic organ prolapse of a quantification stage (POP-Q) ≥2 in the anterior and apical compartments who underwent an anterior sacrospinous ligament fixation and had a minimum follow-up of 6 months after surgery. All patients complained of prolapse symptoms preoperatively.

Two groups with similar characteristics were compared:The native tissue repair group was composed of patients who had an anterior sacrospinous ligament fixation using a prolene suture 0 anchor on the sacrospinous ligament with capio and anterior colporraphy. It was a prospective cohort followed between June 2019 and March 2020.The control group was composed of patients who had an anterior sacrospinous ligament fixation using mesh repair with UHPOLD covering the anterior vaginal wall. It was a historical cohort of patients followed prospectively.

The exclusion criteria were: a POP Q stage < 2 on the anterior compartment and prior treatment for an anterior prolapse with synthetic mesh.

An information letter was presented to inform patients about the study and its modalities.

Due to the nature of the study, no signed informed consent was mandatory. Patients could, at any time, stop participating and oppose the use of their data for research without consequences or prejudice to their medical care.

### 2.3. Procedure

#### 2.3.1. Inclusion Visit

During a preoperative consultation, women had a clinical examination to evaluate the stage of the prolapse according to the standardized POP Q classification. The women were asked to complete the PFDI-20/PFIQ-7/PISQ 12/PGI. They had other examinations prior to surgery: a urinary cytobacteriological analysis, urodynamic assessment if associated urinary symptoms were suspected, and a pelvic ultrasound.

For patients followed prospectively from 2019, the files were selected as follows: a choice of the first 53 patients operated on with ASSLF. The choice of ASSLF was free; the only condition was a significant cystocele (POP Q > 2). In our team, the choice between a vaginal route without mesh or a laparoscopic anterior sacropexy for cystocele was guided by a personalized discussion with the patient.

For patients before 2019 (retrospective cohort), the selection of files was performed as follows: selection of the first 52 files matched with the 53 of the prospective cohort (matched based on age, degree of prolapse, and antecedent cystocele surgery).

#### 2.3.2. Surgical Procedures

The surgical procedures were comparable between the two groups and in line with previously published techniques [13]. The only difference consisted of the use of mesh or a prolene suture plus an anterior colporraphy. The fixation on the sacrospinous ligament was bilateral for all patients.

Before 2019, the most commonly performed vaginal procedures for anterior and apical POP surgical correction in France were mesh surgeries. Vaginal mesh repair procedures are no longer performed in France since the marketing of these implantable devices was stopped by French authorities. Therefore, mesh techniques were preferred before 2019. In our study, patients who received mesh techniques were operated on between 2012 and 2019. After 2019, only native tissue repair techniques could be performed.

Native tissue repair group: The sacrospinous ligaments were needled on each side with a 2.0 Prolene wire, 2 cm medial to the sciatic spine, using the Capio^®^ forceps (Capio Slim™ Boston Scientific, Marlborough, MA, USA). The wires were then attached to the cervix and the uterine isthmus. By pulling on each side of the wire, adjusting the tension on each side, and tightening the knot, the cervix is brought up without excessive tension. To cure the cystocele, an anterior colporrhaphy was performed with bladder plication and/or Halban’s fascial plication.

Mesh repair group: The mesh arm was passed through the sacrospinous ligament on each side using a Capio^®^ type transfixing device 2 cm from the sciatic spine. The prosthesis was fixed to the uterine cervix by two wires previously placed on the lateral edges of the uterine cervix. It was then fixed under the bladder neck by absorbable sutures, allowing for a good spread without folds. One of the two arms were then tensed to raise the cervix.

Associated procedures were sometimes performed: a hysterectomy, mid-urethral sling, and posterior concomitant procedure. The surgical team was composed of three surgeons experienced in vaginal prolapse surgery.

Data were collected on the duration of surgery, length of hospital stay, and any peri-operative complications.

#### 2.3.3. Follow Up

Women had a follow-up visit at 6–8 weeks after surgrey. At this visit, they underwent a clinical examination. Information about any complications or re-interventions was collected. The patients had another visit at 12 months after the initial intervention (an average follow-up of 12 months) during which a clinical examination with a POPQ evaluation was performed. Patients were then asked to complete the PFDI 20/PFIQ7/PISQ 12/ PGI questionnaires. Data were also collected on any complications, re-interventions, or re-admission related to the initial surgery.

### 2.4. Outcomes

Our primary outcome measure was the variation in the position of the Ba point before the intervention versus 12 months after the intervention (at an average follow-up of 12 months). The measurement of the Ba point in cm was evaluated with a ruler by our three referent surgeons and was obtained from the standardized POP Q classification.

Secondary outcome measures were the perception of a bulge in the vagina (a response of “yes” to Question 3 of the PFDI-20 or the Question 8 of the PISQ 12), rates of de novo dyspareunia (response “yes” to the Question 5 of the PISQ 12), and PFDI 20 questions relevant to stress urinary incontinence, urgency, and dysuria (UDI and POP DI). We also collected data on the quality-of-life outcomes (PFIQ7), peri-operative morbidity at the 6-week and 12-month follow-ups, and patient satisfaction using the PGI-I (Patient Global Impression of Improvement) questionnaire or another type of Likert scale at the end of the study.

### 2.5. Statistical Analyses

The primary hypothesis was that the anterior colporraphy without mesh was less performant for anterior anatomical result compared with the mesh repair for cystocele when performed at the same time as ASSLF.

Based on the results available in the Cochrane 2013 and, in particular, in the El Nazer study of the anterior pelvic compartment, which compared anterior colporrhaphy versus prosthetic reinforcement with a first-order risk of 5%, a power of 90%, a conservative standard deviation of 1.4, and expected means of −1.8 and −2.7 [8,11], our study would need to include 51 patients in each group.

Our study was designed to compare two groups with similar characteristics that were matched on age, preoperative previous stage, preoperative sexual activity, and a history of surgery for prolapse.

A descriptive analysis was performed on the patients’ characteristics. Mean values with standard deviation and/or medians and quartiles were given for quantitative variables, while frequencies and proportions were described for categorical variables. Qualitative variables were compared with the chi-square test or Fisher’s exact test when appropriate. Quantitative variables were compared with the non-parametric Wilcoxon rank-sum test. The level of statistical significance was set at 0.05. A multivariate analysis was implemented using a logistic model or ANCOVA model. For the logistic model, variables were selected with a stepwise procedure, with an alpha-to-enter of 0.15 and an alpha-to-exit of 0.05. Statistical analyses were performed using SAS software, version 9.4 (SAS Institute, Cary, NC, USA).

## 3. Results

Fifty-three patients (the prospective cohort) were included in the native tissue repair group. To compose the mesh group, 272 out of 359 mesh patients were pre-selected to be matched (follow-ups, selection variables filled in), and 52 patients were selected to match based on age, preoperative previous stage, preoperative sexual activity, and history of surgery for prolapse (ratio 1:1).

The participants’ characteristics are shown in Table 1. The two groups were comparable. The mean age was 71.22 (range 53.70–85.33). Most patients were not obese and did not have diabetes mellitus. The majority of patients had no previous hysterectomy and no previous surgery for POP or urinary incontinence. The median follow-up was more than 1 year (range 0.5–1.71 year). The majority of patients reported no sexual activity (single 35.71%, due to prolapse 20%, others 35.71%). The majority of those who reported sexual activity did not suffer from dyspareunia. Two groups were not comparable on the C point preoperatively. Posterior colporraphies were performed more often in the native tissue repair group.

The primary and secondary outcomes are presented in Table 2.

For the primary endpoint, there was no difference between the two groups on the Ba point at the 12-month follow up (*p* = 0.9789). The apex was significantly better corrected in the native tissue repair group (−7 vs. −6 *p* = 0.0007). There was also a better correction on the rectocele in the native tissue repair group (−3 vs. −2, *p* = 0.0178). For the anterior wall at 12 months, Stage 2 was recorded for 32.69% in the native tissue repair group and 26.92% in the mesh group, with no significant difference.

A total of 35 women reported being sexually active at the 12-month follow-up. Eight women who were not sexually active before surgery (six due to prolapse, one due to a male problem, and one due to another reason) became sexually active after surgery. Seven women who were sexually active before surgery reported no sexual activity after surgery (four because it was not desired, two due to a male problem, one due to apprehension). Among the 35 sexually active patients after surgery, 15 described dyspareunia “sometimes” or “rarely”, with no statistical difference between the two groups (35.71% versus 58.82%, *p* = 0.2852). None of the patients had severe dyspareunia. After analysis, none of those who were sexually active after surgery suffered from de novo dyspareunia.

Most patients reported feeling better or much better on the PGI-I or other Likert scale (97.5% in the native tissue repair group and 97.2% the in mesh group). The analysis of the PFDI-20 global score showed a significant difference between the native tissue repair and mesh group (median score: 27.08 vs. 18.75, *p* = 0.0360), to the disadvantage of the native tissue repair group. However, a high number of missing data must be considered. Investigating the details of the key stand-alone questions of the PFDI-20 questionnaire, the questions demonstrate that stress incontinence and urge incontinence were not significantly different between the two groups. For both groups, the median PFIQ7 score was close to 0 (indicating a good quality of life regarding prolapse symptoms).

For colorectal symptoms (CRADI), we found a statistical difference between the groups at the one-year follow-up (a better result in the mesh group, mean 15.39 vs. 7.72), but the two groups were not matched on colorectal symptoms.

The multivariate analyses performed to search for predictive factors for the anatomical results at 1 year after surgery (Ba, C, and Bp points) were inconclusive and will not be presented.

The perioperative morbidity and complications are described in Table 3.

In native tissue repair group, the length of stay was shorter and the operative time was longer. There was no difference between the two groups for complication rates. There was no difference between the two groups for mid-urethral sling after prolapse surgery (five in the native tissue repair group and six in the mesh group (*p* = 1.000)).

## 4. Discussion

It is important to note that all patients included in our study were patients with a significant degree of prolapse and with no severe preoperative dyspareunia or urinary incontinence. They represented the general patient population consulting in our uro-gynecological center.

Contrary to the initial hypothesis, this prospective comparative study showed similar anatomical results with respect to the Ba point at 12 months after native tissue repair or mesh reinforcement for the cystocele when performed concomitantly with AFFSL.

The apex was significantly better corrected in the native tissue repair group, but the two groups were not matched on this point preoperatively.

There was also a better correction on the rectocele in the native tissue repair group. A multivariate analysis was performed and did not find predictive factors to explain this difference (associated autologous posterior repair does not affect the results).

The native tissue repair group had a shorter length of stay. This difference might be due to our organization and not to the type of surgery performed. Indeed, since 2016, our outpatient service has been improved and active early rehabilitation increased. The mesh group was treated in 2009, whereas the native tissue repair group was treated in 2019.

The native tissue repair group also had a longer operative time. This can be explained by more associated procedures and the learning curve in progress for this technique. When operating on the mesh group, the surgeons in the study had already been performing ASSLF procedures for two years. However, the surgeons operating on the native tissue repair group in 2019 had less experience with this technique as a result of the sudden cessation of the mesh.

We found that the rate of postoperative complications, de novo stress urinary incontinence, reoperation for recurrence, and exposure was similar in both groups, whereas we would have expected a higher rate of re-intervention for the mid-urethral sling or exposure in the mesh group.

We found no de novo dyspareunia in sexually active women after surgery. We found similar rates of dyspareunia in our mesh group compared to other studies [14].

A Cochrane review of results from ten trials in 558 women (published in 2016) comparing native tissue repair versus any transvaginal permanent mesh reported a better correction on the Ba point after surgery in the mesh group (mean difference −0.93 [−1.27, −0.59], *p* < 0.0001) [10]. It is important to note that in these studies, native tissue repair consisted of an anterior colporraphy without an anterior apical suspension on the sacrospinous ligament. A prospective multicenter study comparing anterior colporraphy versus transvaginal mesh showed a significant difference in the Ba point at a 7-year follow-up (Ba ≥ −1: 46% vs. 3%, *p* = 0.0036). The primary endpoint in functional results was similar at this long-term follow-up [5]. These studies all compared cystocele repair with or without mesh, but did not evaluate apical correction (native tissue repair consists only of the colporraphy anterior without an associated ligament fixation). Although anterior compartment prolapse appears to be the most prevalent type of POP, most high-grade cystoceles are associated with concomitant apical support loss.

We hypothesized that apical suspension on the SSL could worsen the results on the anterior wall because of the new direction of the vagina. The direction of the vagina seems to be slightly different if the sacrospinous fixation of the apex is performed via the anterior approach or posterior approach. Goldberg showed a better anterior result when apical fixation was performed by the anterior approach compared to the posterior approach with systematic anterior colporraphy in both groups (Ba −2.50 in ASSLF vs. −1.61 in the posterior approach, *p* = 0.005) [12].

In their observational study, Bastani and al. compared anterior and posterior approaches for the apex correction. They did not analyze the anatomical outcome, but showed a statistically significant difference in the CRADI-8 subscale (*p* < 0.001) and the total score of PFDI-20 (*p* = 0.047) in favor of the anterior approach (an associated colporraphy anterior and posterior was performed in both groups). Surprisingly, we also found significant improvement of the CRADI subscale in the meshless group; however, the physiopathological explanation for this finding is unclear [15].

Although the anterior SSLF approach is used less frequently, this procedure is appropriate if posterior compartment prolapse repair is not indicated and if the treatment of a cystocele is required.

An observational study by Petruzelli et al. was the first to demonstrate that treatment with a combination of sacrospinous fixation and an anterior colporraphy procedure using a single anterior incision was efficient on the anterior and apical stages (1/42 unsuccessful treatments leading to a reoperation before 6 months after the primary surgery) [16].

A recent feasibility study was published in 2021 on the ASSLF technique combined with anterior colporraphy, finding good clinical and functional results (Ba −2.5 before surgery, −3 after surgery; *p* < 0.001; median follow-up at 10 months). The technique was feasible, with rare intraoperative morbidity and a low recurrence at short-term follow-up [17]. Another study compared two groups treated with ASSLF: one with the classic ASSLF technique (anterior fixation on sacrospinous ligament) and the other with sacrospinous ligament fixation plus ischial spinal fascia. Both groups had a concomitant hysterectomy and anterior and posterior colporraphy. In terms of pelvic floor function, the study showed less anterior and apical recurrence at 3 months and 6 months after surgery (Stage 2: 4.55%) than the other studies on native tissue repair. The anatomical result was not different in the two groups [18].

To date, no study has compared native tissue repair to mesh repair on the correction of cystocele when the procedure is associated with anterior apical ligament fixation.

Since the use of vaginal mesh has been discontinued in France, the only way to compare, prospectively, mesh and meshless anterior and apical suspension is to compare laparoscopic anterior sacropexy and ASSLF + vaginal anterior colporraphy. To date, we do not have any data comparing ASSLF + vaginal anterior colporraphy versus anterior sacrocolpopexy. Future prospective randomized trials could be performed.

There is little data regarding results after a sacrocolpopexy compared to anterior and apical vaginal mesh. In a multicenter randomized controlled trial, Lucot and al. compared two mesh reinforcement techniques by two different surgical routes in the treatment of concomitant cystocele and apex. After 1 year, there was no difference in the Ba point (−1.3 in the laparoscopic sacropexy group vs. −4 in the vaginal mesh group; *p* = 0.621) [19]. De Tayrac and al. showed the same results in a prospective, multicenter, randomized open-label study with two parallel groups treated by sacrocolpopexy or Uphold Lite vaginal mesh associated with anterior colporraphy (Ba point −3 in each group; *p* = 0.73) [14].

Therefore, it is important to note the strengths of this study. The technique described has made it possible to compensate for the cessation of mesh us, and to cure cystocele and apex at the same time. It is easy to perform for a trained surgeon, and there was no increase in the incidence of major complications. In this study, surgeries could be realized by three different surgeons. In fact, the surgical team was composed of three surgeons experienced in vaginal prolapse surgery with the same formation. The postoperative evaluation was performed by the operator, but the method of physical examination was the same between surgeons and was based on the universal POP Q classification. We used validated questionnaires to evaluate objective outcomes. We enrolled a sufficient number of subjects to establish the difference of 1 cm described in our starting hypothesis.

The study’s main limitations are as follows:

The control group was followed retrospectively, and we could not recover the missing data, especially from the questionnaire. The median follow-up was at approximately 1 year, which is relatively short, although we do know that most recurrences and complications occur within the first year [20]. In this study, we did not assess the risk of post operative de novo rectocele; this might be interesting to do in order to compare the de novo prolapse rate in the untreated vaginal compartments following vaginal native tissue repair and vaginal mesh repair. It would be interesting to evaluate the results in long-term studies, despite the inevitable high dropout rates in subsequent follow-up, especially among patients satisfied at one year and among the elderly.

## 5. Conclusions

This study found comparable anatomical results on the Ba point between mesh and meshless ASSLF at a 12-month follow-up. A multivariate analysis did not find confounding factors to explain the significant difference between the two groups on the C and Bp points at 12 months. These results are encouraging for the development of this native tissue repair technique (=ASSF), which is designed to treat cystoceles and apical defects in the same approach. Further prospective comparisons of this native tissue repair technique with other mesh suspension (vaginal or laparoscopic) are necessary to confirm these preliminary findings.

## Figures and Tables

**Table 1 jcm-12-02212-t001:** Participant characteristics (N = 104).

	Native Tissue Repair N = 52	Mesh N = 52	*p* Value (Fisher Wilcoxon)
Age (year), median (Min; Max)	71.22 (53.7; 85.33)	70.78 (54.97; 82.25)	0.8940
BMI (kg/m^2^), median (Min; Max)	23.34 (16.14; 34.16)	24.24 (18; 34.89)	0.4286
POP Anterior stage pre-operative >2	48 (92.31%)	48 (92.31%)	1.0000
Previous surgery for POP without mesh	2 (3.85%)	2 (3.85%)	1.0000
Time between surgery and visit at 1 year median (Min; Max)	1.06 (0.5; 1.71)	1.03 (0.52; 1.43)	0.8708
Diabetes	6 (11.54%)	3 (5.77%)	0.4878
Smoking status	2 (3.85%)	2 (4%)	1.0000
Parity median (Min; Max)	2 (0; 5)	3 (1; 4)	0.9879
Previous hysterectomy	4 (7.69%)	1 (1.92%)	0.3627
Reason for no sexual activity		DM 6	
No partner	10 (28.57%)	15 (42.86%)	
Due to prolapse	8 (22.86%)	6 (17.14%)	
Other reason	17 (48.57%)	8 (22.86%)	
Previous surgery for urinary incontinence	2 (3.85%)	4 (7.69%)	0.6781
Sexual activity (yes)	17 (32.69%)	17 (32.69%)	1.0000
Sexual activity with dyspareunia	6/17 (66.67%)	12/17 (70.59%)	1.0000
PISQ 12 Score (/100), median (Min; Max)	38.18 (22.91; 48)	31 (17; 39)	0.0581
Ba point, preoperative median (Min; Max)	3 (0; 7)	3.8 (−1; 8)	0.1550
C point, preoperative median (Min; Max)	0 (−6; 7)	0.8 (−4; 9)	0.0256
Bp point, preoperative median (Min; Max)	−1.5 (−3; 5)	−1 (−3; 7)	0.2363
Anterior colporraphy (yes)	52 (100%)	48 (92.31%)	0.1178
Bladder plication	33 (63.46%)	1 (2.13%) MD 5	1.0000
Posterior colporraphy (yes)	36 (69.23%)	11 (21.15%)	<0.0001
Mid-urethral sling (yes)	12 (23.08%)	10 (19.23%)	0.8107
Concomittant hysterectomy (yes)	3 (5.77%)	13 (25%)	0.0125

Data are medians [25th; 75th percentiles] or numbers and percentages. BMI—body mass index, POP—pelvic organ prolapse, PISQ-12—Pelvic Organ Prolapse/Urinary Incontinence Sexual Questionnaire.

**Table 2 jcm-12-02212-t002:** Postoperative anatomical and functional results.

	Native Tissue Repair N = 52	Mesh N = 52	*p* Value (Wilcoxon)
**Anatomical**			
Ba point at 12 months, median (Min; Max)	−2 (−3; 1.5)	−2 (−3; 1)	0.9789
C point at 12 months, median (Min; Max)	−7 (−9; −3)	−6 (−8; 1)	0.0007
Bp point at 12 months, median (Min; Max)	−3 (−3; 0)	−2 (−3; 0.5)	0.0178
Anterior stage POP Q at 12 months			
1	35 (67.31%)	38 (73.08%)	0.6685
≥2	17 (32.69%)	14 (26.92%)	
**Sexual**			
Sexual activity at 12 months surgery (yes)	18 (35.29%)	17 (32.69%)	0.8369
No sexual activity pre operative -> Sexual activity post operative	5	3	-
Sexual activity pre operative -> No sexual activity post operative	4	3	-
Reasons for no sexual activity post operative	4 No desired	2 male problem 1 apprehension	-
PISQ Score(/100) at 12 months, median	37 (24; 44)	37 (30; 43)	0.8666
Yes to Q5 PISQ 12: Do you feel pain during sexual intercourse?	5 (35.71%)	10 (58.82%)	0.2852
Yes to Q8 PISQ 12: Do you avoid intercourse because of prolapse?	1 (9.09%)	1 (5.88%)	1.0000
**Qol**			
PGI better or much better at 12 months, n (%)	39/40 (97.5%) MD 12	35/36 (97.2%) MD 16	-
PFDI 20 at 12 months, total score/300	27.08 (0; 108.33) MD 14	18.75 (0; 80.21) MD 4	0.0360
Sub-score UDI-6 at 12 months, median	8.33 (0; 66.67)	4.17 (0; 41.67)	0.2868
Sub-score POPDI-6 at 12 months, median	8.33 (0; 41.67)	0 (0; 29.17)	0.0622
Sub-score CRADI-8 at 12 months, median	6.25 (0; 43.75)	6.25 (0; 28.13)	0.0105
Yes to Q3 PFDI 20: At 12 months, do you often feel like you can’t empty your bladder?	4 (10%)	1 (2%)	0.1672
Yes to Q16 PFDI 20: At 12 months, do you have frequent involuntary leakage associated with the urge to urinate?	13 (32.5%)	11 (22%)	0.3387
Yes to Q17 PFDI 20: At 12 months, do you often leak urine when you cough, sneeze, or laugh	10 (25%)	11 (22.45%)	0.8064
PFIQ 7 at 12 months, total score/300, median	0 (0; 100)	0 (0; 38)	0.6574

Data are medians [25th; 75th percentiles], or numbers and percentages. MD—missing data, POP-Q—pelvic organ prolapse quantification system, PFIQ—Pelvic Floor Impact Questionnaire, PFDI—Pelvic Floor Distress Inventory, UDI6—Urinary Distress Inventory, CRADI8—ColoRectal Distress Inventory, POPDI6—Pelvic Organ Prolapse Distress Inventory, PISQ-12—Pelvic Organ Prolapse/Urinary Incontinence Sexual Questionnaire, Qol—Quality of life.

**Table 3 jcm-12-02212-t003:** Perioperative complications and reoperations.

	Native Tissue Repair = 52	Mesh N = 52	*p* Value (Fisher)
Perioperatively, n	9/52 (2 hematoma, 5 urinary retention: 2 with self urinary catheterization, 1 with strip lowering, 2 cystisis)	4/52 (1 hematoma, 2 urinary retention: 1 with hetero urinary catheterization, 1 cystitis)	0.2350
Operative time (min), median	65.5 (22; 200)	55 (25; 144)	0.0486
Length of stay (days), median	1 (1; 7)	3 (1; 7)	<0.0001
Pain < 3 weeks, n	10/52	3/52	0.0721
Urinary retention at 6 weeks, n	0	0	-
Vaginal or visceral mesh or prolene exposition at 12 months, n	0	0	-
Reoperation for recidive at 12 months, n	1 sacrocolpopexy posterior	2: hysterectomy for cervical elongation, sacrocolpopexy posterior	
Reoperation for complication at 6 weeks and 12 months, n	0	0	-
Mid-urethral sling post intervention	5	6	1.0000

## Data Availability

Not applicable.

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
