# Peer review of "Anterior and Apical Prolapse: Comparison of Vaginal Mesh Surgery to Vaginal Surgery with No Mesh"

_jcm, 2023, doi:10.3390/jcm12062212_

Round 1

Reviewer 1 Report (Previous Reviewer 3)

Thank you for your answers. 

1. Line 52 : Before 2019, the most commonly performed vaginal procedures for anterior and 51 apical POP surgical correction in France were mesh surgeries. This native tissue repair 52 technique reduced the risk of prolapse recurrence, shorter operating time and cost bene-53 fits. (5–7) But, after evaluations of mesh-related complications (vaginal mesh exposure, 54 chronic pain and dyspareunia), the FDA decided to stop the sale and production of im-55 plantable devices marketed for vaginal prolapse surgery in the USA.

Please clarify 

2. Explain how the choice of native tissue or mesh technics was realized for each patients : clinical data, choice of patients, choice of surgeon

It must be added in the methods section 

3. 7. Does surgery realizeds by several surgeon for each technic or the same one?

It must be added in methods section and then discuss in the discussion part. 

Author Response

You asked us for minor corrections. We have made these corrections. Below you will find the answers to your questions, for which we have also made the changes in the article :

  1. Line 52 : Before 2019, the most commonly performed vaginal procedures for anterior and apical POP surgical correction in France were mesh surgeries. This native tissue repair technique reduced the risk of prolapse recurrence, shorter operating time and cost benefits. (5–7) But, after evaluations of mesh-related complications (vaginal mesh exposure, chronic pain and dyspareunia), the FDA decided to stop the sale and production of implantable devices marketed for vaginal prolapse surgery in the USA.

Before 2019, the most prevalent vaginal procedure for the correction of anterior and apical prolapse used a reinforcement with synthetic mesh. The use of these implants has reduced anatomical recurrence rates, but it simultaneously introduced graft-related complications. Because of these complications, the FDA has issued warnings about the use of these mesh. As a result of these warnings, the production of these mesh has been stopped in the United States and the use of these materials was stopped in several countries, including France.

Since 2019, surgeons in countries where vaginal mesh repairs are no longer options have returned to older surgical techniques using native tissue repair for correction of high grade cystocele and apical defect.

In our staff we choose the option of anterior suspension of the apex on the sacrouspinous ligament (the same route as anterior graft route) without mesh, just with one or two stitches. We designed a new technique without mesh.

  1. Explain how the choice of native tissue or mesh technics was realized for each patient: clinical data, choice of patients, choice of surgeon.

It must be added in the methods section

First we want to precise that our study was not randomized. We performed a monocentric prospective study comparing two similar cohorts of women.

For patients followed prospectively from 2019, the files were selected as follows: choice of the first 53 patients operated with ASSLF. The choice of ASSLF was free, the only condition was a significant cystocele (POP Q > 2). In our team, the choice between a vaginal route without mesh or a laparoscopic anterior sacropexy for cystocele is guided by a personalized discussion with the patient. The study population consisted of women older than 18 years old with pelvic organ prolapse quantification stage (POP-Q) ≥ 2 in the anterior and apical compartments, who had an anterior sacrospinous ligament fixation and a minimum follow-up of 6 months after surgery. All patients complained of prolapse symptoms preoperatively.

For patients before 2019 (retrospective cohort), the selection of files was done as follows: selection of the first 52 files matched with the 53 of the prospective cohort (matching on age, degree of prolapse and atcd of cystocele surgery).

  1. 7. Does surgery realizeds by several surgeon for each technic or the same one?

It must be added in methods section and then discuss in the discussion part.

In this study, surgeries could be realizing by 3 differents surgeons. In fact, surgical team composed of 3 surgeons experienced in vaginal prolapse surgery, with the same formation. The postoperative evaluation was done by the operator but the method of physical examination was the same between surgeons and based on the universal POP Q classification.

Reviewer 2 Report (Previous Reviewer 2)

The language/content has been improved, as indicated. I have no further remarks considering this manuscript, which may be published in a present form.

Author Response

Thank you for your approval

Reviewer 3 Report (New Reviewer)

This is a well-designed prospective study investigating the efficacy and safety of two surgical vaginal procedures for anterior and apical POPwith a clear and  clinical message regarding the surgical approach,adequte  instruction and methodology with informative and good quality tablesand well designed statistical analysis.

Author Response

Thank you for your approval

Reviewer 4 Report (New Reviewer)

I precisely reviewed a manuscript entitled “ANTERIOR AND APICAL PROLAPSE: comparison of vaginal mesh surgery to vaginal no mesh surgery”.  Since 2019, surgeons practicing in countries where vaginal mesh repairs are no longer options have been worried about the complications (cystocele failure or recurrence) associated with reverting to the vaginal approach for high grade cystocele and apical defect.  To date, no study has compared native tissue repair to mesh repair on the correction of cystocele, when the procedure was associated with anterior apical ligament fixation.  For this reason, this study is valuable for urogynecologists for high grade cystocele and apical defect.  Study design is also well organized and manuscript is well written.

Author Response

Thank you for your approval

Reviewer 5 Report (New Reviewer)

introduction is too long; about the results denovo   urge and stress were evaluated?in discussion  is important to underline also the denovo prolapse in posterior vaginal wall,in these sense these twio paper could helkp for the discussion

  • DOI: 10.1111/j.1471-0528.2011.03231.x

doi: 10.3109/01443615.2015.1086990. Epub 2015 Oct 22.c

Author Response

introduction is too long :

We have modified the introduction

about the results denovo   urge and stress were evaluated?

No we just evaluate the pourcentage of  urge and stress incontinence at 12 months follow up after surgery (results presented in the table 2). But it’s not evaluate if these patients had the same symptoms before surgery.

in discussion  is important to underline also the denovo prolapse in posterior vaginal wall,in these sense these twio paper could helkp for the discussion :

In this study we did not assess the risk of post operative de novo rectocele, it was not included in our primary or secondary objectives. It might be interesting to do, in order to compare the de novo prolapse rate in the untreated vaginal compartments following vaginal native tissue repair and vaginal mesh repair.

We have included this remark in the limits of the study.  

This manuscript is a resubmission of an earlier submission. The following is a list of the peer review reports and author responses from that submission.

Round 1

Reviewer 1 Report

The study includes a small sampling from participants  I do not understand the reason why  from a total satisfactory number of participants only a small  part was evaluated in the study

Reviewer 2 Report

In the paper, the Authors describe clinical results of a native tissue anterior sacro-spinal vaginal suspension, as compared to anterior sacro-spinal repair using mesh. The study design, the patients groups and the methods are well selected, and reflect present clinical needs. They focus on the improvement in patients with pelvic organ prolapse.

Surgical procedures are clearly presented and described, enabling a comparison to surgical approach used in other centers to cure prolapse. The subject is actual, and world-wide recognized. This is an important study showing the effectiveness of a prolapse treatment using native tissues, as opposed to the broadly banned treatment with mesh. 

I recommend publishig this paper to the benefit of readers, and particularly to the benefit of our patient, after some style improvements. I also suggest using "Past Tense" English throughout the whole text. The paper might profit, if the language were consulted by a native-speaker.

Reviewer 3 Report

Original study comparing native technic versus mesh for prolapse management.

This article deserve to publish after major revisions.

The article must be edited by english native speaker.

Abstract :

1) line 18 : change subjective by objective results

2) line 21 and throughout the text : explicit how apex and rectocele were  assessed ?

Introduction

3) It must be shorten

4) line 52 : "native tissue repair? ". Do you mean technic with mesh or without mesh?

5) line 87 : the objective is to compare to technic and not to show that one is inferior than the other. Please explain your hypothesis in the material and methods in statistical analysis.

Matériel and Methods

6) Explain how the choice of native tissue or mesh technics was realized for each patients : clinical data, choice of patients, choice of surgeon

7) Which kind of methods was used to match patients ?

8) line 184 : Do you mean ANOVA ?

9) Does surgery realizeds by several surgeon for each technic or the same one?

Discussion

10) It must be rewritten in a way to emphasize the main results.

11) Can you explain that the rate of stage 2 anterior vaginal prolapse at one 23 year was about 30% in both groups?